# The Anticancer Potential of *Psidium guajava* (Guava) Extracts

**DOI:** 10.3390/life13020346

**Published:** 2023-01-28

**Authors:** Bronwyn Lok, Dinesh Babu, Yasser Tabana, Saad Sabbar Dahham, Mowaffaq Adam Ahmed Adam, Khaled Barakat, Doblin Sandai

**Affiliations:** 1Department of Biomedical Science, Advanced Medical and Dental Institute, Universiti Sains Malaysia, Kepala Batas 13200, Penang, Malaysia; 2Faculty of Pharmacy and Pharmaceutical Sciences, University of Alberta, Edmonton, AB T6G 2E1, Canada; 3Department of Science, University of Technology and Applied Sciences Rustaq, Rustaq PC 329, Oman; 4Department of Chemistry and Biochemistry, San Diego State University, San Diego, CA 92182, USA

**Keywords:** *Psidium guajava*, cancer, cytotoxicity, flavonoids

## Abstract

The fruits, leaves, and bark of the guava (*Psidium guajava*) tree have traditionally been used to treat a myriad of ailments, especially in the tropical and subtropical regions. The various parts of the plant have been shown to exhibit medicinal properties, such as antimicrobial, antioxidant, anti-inflammatory, and antidiabetic activities. Recent studies have shown that the bioactive phytochemicals of several parts of the *P. guajava* plant exhibit anticancer activity. This review aims to present a concise summary of the in vitro and in vivo studies investigating the anticancer activity of the plant against various human cancer cell lines and animal models, including the identified phytochemicals that contributes to their activity via the different mechanisms. In vitro growth and cell viability studies, such as the 3-(4,5-dimethylthiazol-2-yl)-2,5-diphenyltetrazolium bromide (MTT) assay, the sulforhodamine B (SRB) assay, and the trypan blue exclusion test, were conducted using *P. guajava* extracts and their biomolecules to assess their effects on human cancer cell lines. Numerous studies have showcased that the *P. guajava* plant and its bioactive molecules, especially those extracted from its leaves, selectively suppress the growth of human cancer cells without cytotoxicity against the normal cells. This review presents the potential of the extracts of *P. guajava* and the bioactive molecules derived from it, to be utilized as a feasible alternative or adjuvant treatment for human cancers. The availability of the plant also contributes towards its viability as a cancer treatment in developing countries.

## 1. Introduction

*Psidium guajava*, commonly known as the guava plant, is a small evergreen shrub belonging to the Myrtaceae family [1]. The Caribbean and Central and South America are the native origin of this plant [2], but it has now been widely cultivated around the world, particularly in the tropical and subtropical regions for its fruit [3]. The *P. guajava* plant is a small tree that grows nearly up to 6 m in height, branching out close to the ground with large thin flakes of bark that peel off revealing a smooth, green layer beneath (Figure 1a). The leaves are oval or oblong-elliptic-shaped, with the bottom surface having prominent veins growing in an opposite arrangement on the stems, and are aromatic when crushed (Figure 1b). The plant bears faintly fragrant white flowers in singles or clusters with four or five petals and a prominent tuft of white stamens. The fruits range in size and shape (round, ovoid, or pear-shaped, with thin, light-yellow or green skin and many small, hard seeds (Figure 2a)). The flesh of the fruit is granular and can either be white, pink, or red (Figure 2b) [1,4].

The leaves, fruits, barks, and roots of the *P. guajava* plant have been traditionally used in Africa, Asia, and South America for treating various ailments, either as a topical application or orally as a decoction, infusion, and boiled preparation. Due to their antibacterial and analgesic properties, these preparations were used to treat diabetes mellitus, hypertension, obesity, rheumatism, and gastrointestinal diseases like diarrhea and stomachache [5]. The medicinal properties of the leaves and fruits of the *P. guajava* plant can be attributed to the high amount of carotenoids, essential oils, flavonoids, phenolic compounds, and vitamins [6].

Recently, the World Health Organization (WHO) has highlighted that cancer is the second leading cause of death globally, with a prediction of 9.6 million deaths in 2018 [7]. With the growing global cancer burden, approximately 70% of cancer deaths occur in low- and middle-income countries, where the health systems of these countries are the least prepared to manage this alarming health risk [7]. Therefore, the development of economical cancer treatment solutions is urgently needed. Nature has provided us with still untappable resources with potential health benefits. Among them, various parts of *P. guajava* have been recently gaining attention for their anticancer properties [8,9,10].

Numerous in vitro studies have been performed to assess the anticancer potential of different parts of the *P. guajava* plant. The anticancer potential of an extract is usually evaluated using in vitro assays, such as the 3-(4,5-dimethylthiazol-2-yl)-2,5-diphenyltetrazolium bromide (MTT) assay, 3-(4,5-dimethylthiazol-2-yl)-5-(3-carboxymethoxyphenyl)-2-(4-sulfopheny)-2H-tetrazolium (MTS) assay, the sulforhodamine B (SRB) assay, thymidine incorporation assay, and the trypan blue exclusion test, to assess their cytotoxicity towards cancer cell lines. The MTT assay is a colorimetric assay commonly used to assess metabolic activity, which reflects the number of metabolically active and viable cells present. MTT is a yellow dye that is readily taken up by viable cells due to its water-soluble nature, which gets reduced into the purple-colored formazan in the mitochondria. The formazan crystals formed would be dissolved using an organic solvent like dimethyl sulfoxide; then, the number of viable cells can be measured using a spectrophotometer based on the amount of formazan formed [11]. The MTS assay is similar to the MTT assay but with tetrazolium salts that can be reduced by viable cells resulting in the generation of formazan products directly soluble in a cell culture medium [12]. This property of the tetrazolium salts eliminates the media removal and dissolution in the organic solvent, which increases the assay accuracy compared to MTT.

The SRB assay is another colorimetric assay used to determine cell density based on the measurement of cellular protein content. The fluorescent dye SRB binds to the proteins of the cells under mild acidic conditions. After the excess dye gets washed away, the amount of dye taken up by the cells serves as a proxy for cell mass and thus, the number of cells in a sample can be measured using a microplate reader [13]. In addition to the SRB assay, the neutral red cell cytotoxicity assay is another commonly used colorimetric assay to detect cell viability or drug cytotoxicity. Neutral red is a eurhodin dye that can be taken up by viable cells and gets incorporated into their lysosome. The incorporated dye can be released from the viable cells in an acidified ethanolic solution and the amount of released dye serves as a quantitative measurement of the number of viable cells at an optical density (OD) of 540 nm [14].

Another reliable way to evaluate cell proliferation is by quantitatively measuring cellular DNA synthesis. The thymidine incorporation assay is based on the concept of utilizing the measurement of the radioactive nucleoside, 3H-thymidine, into new strands of chromosomal DNA during mitotic cell division [15]. The trypan blue exclusion test is one of the earliest methods for measuring cell viability [16]. Trypan blue is a dye that will not be absorbed by live cells with intact cell membranes due to their membrane’s selective permeability. As live cells do not take up the dye, the number of viable cells and dead cells with the stained dye can be distinguished and quantified under a microscope [16].

Various chromatographic techniques were utilized to isolate the active biomolecules from the *P. guajava* plant extracts, such as column chromatography, high-performance liquid chromatography (HPLC), and thin layer chromatography (TLC). The isolated biomolecules were then characterized and identified via spectral analyses that include time-of-flight mass spectrometry (TOFMS), Fourier-transform infrared spectroscopy (FTIR), and nuclear magnetic resonance (NMR) spectroscopy [17,18]. Some studies also supplemented the NMR spectroscopy with heteronuclear single quantum coherence (HSQC) and heteronuclear multiple bond correlation (HMBC) experiments to obtain further details on the structure and dynamics of proteins [18,19].

Upon a thorough literature search on PubMed (between 2004 and 2022), a total of 43 different in vitro studies were conducted on various human cancer cell lines using extracts from the leaves, fruits, seeds, barks, branches, and roots of the *P. guajava* plant. For some of those extracts, the active components contributing towards their anticancer activity have been identified, either through their contribution towards/against cancer cell apoptosis, angiogenesis, and metastasis. In addition, a couple of in vivo studies using the mice model have been reported that evaluate the effects of these extracts on solid tumors.

## 2. Anticancer Effects of the *Psidium guajava* Plant

A list of published studies reporting the in vitro and in vivo anticancer activities from the various extracts derived from the different parts of the *P. guajava* plant, along with the bioactive molecules that are associated with their anticancer mechanisms against human cancer cell lines are summarized in Table 1, Table 2 and Table 3, respectively. Figure 3 outlines the various mechanisms and biomolecules involved in the anticancer effects of *P. guajava* plant extracts.

### 2.1. Anticancer Effects of Psidium guajava Leaf Extracts

An in vitro and in vivo anticancer study of dichloromethane and ethanolic crude extracts of *P. guajava* leaves, as well as their fractions, were conducted on nine cancer cell lines—the NCI-H460 lung cancer, MCF-7 breast cancer, HT-29 colon cancer, PC-3 prostate cancer, K562 leukemia, 786-0 kidney cancer, OVCAR-3 ovarian cancer, NCI/ADR-RES resistant ovarian cancer, and UACC-62 melanoma cell lines [8]. The antiproliferative effects of the extracts on these cancer cell lines were investigated using the SRB assay. The dichloromethane extract had a more potent antiproliferative activity than the ethanolic extract. Therefore, the dichloromethane extract was subsequently fractionated, with the fractions evaluated on the cancer cell lines. The crude dichloromethane extract had a good effect against the cancer cell lines, with total growth inhibition (TGI) values (the necessary concentration for total inhibition of cancer cell proliferation) of 26, 29, 32, 43, 44, 49, 61, 64, and 65 μg/mL against the OVCAR-3, PC-3, K562, HT-29, 786-0, NCI/ADR-RES, MCF-7, NCI-H460, and UACC62 cells, respectively. Among the various fractions tested, the fraction designated as PG.(F2-F3).(F6-F9).(F4) was found to be the most active fraction in terms of the activity against the cancer cell lines, with TGI values ranging from 1 μg/mL for the OVCAR-3 cells to 37 μg/mL for the UACC62 cells. The TGI values of the fraction against the K562, NCI/ADR-RES, NCI-H460, HT-29, MCF7, PC-3, and 786-0 cells were 2, 4, 5, 5, 8, 12, and 28 μg/mL, respectively. Phytochemical purification of the fraction revealed the presence of 4 isomers meroterpenes: guajadial, psidial A, and psiguadials A and B (Figure 4) [2]. Guajadial, a meroterpenoid isolated from *P. guajava* leaves, was shown to exhibit antineoplastic activity against non-small-cell lung carcinoma, inhibiting the proliferation and migration of the A549 and H1650 lung cancer cell lines. The research also showed that guajadial suppresses A549 tumor growth in xenograft mice [57]. From the crude dichloromethane guava leaf extract by Rizzo et al., an enriched guajadial fraction was obtained via sequential chromatography. The guajadial fraction was shown to have significant selective antiproliferative activity against MCF-7 and MCF-7 BUS breast cancer cells, with TGI values of 5.59 and 2.27 µg/mL, respectively, as assessed by the SRB assay [20].

Four unusual meroterpenoids were isolated from a methanolic extract of guava leaves by Gao et al. The cytotoxicity of the isolated compounds guajadial C, D, E, and F were investigated on the A549, MCF-7, HL60, and SMMC-7721 cells (a derivative of the cervical cancer HeLa cells) using the MTS assay with minor modifications. All the compounds investigated showed inhibitory effects against the five human cancer cell lines, except that guajadial C and D showed no effect on the growth of MCF-7 cells. Guajadial E especially exhibited a strong inhibitory effect against the cancer cells, with half-maximal inhibitory concentration (IC_50_) values of 6.30, 7.78, 7.77, and 5.59 μg/mL against the A549, MCF7, HL60, and SMMC-7721 cells, respectively [18].

From a 75% ethanolic extract of *P. guajava* leaves, guavinoside C (avicularin) (Figure 4), guavinoside F, and quercetin were isolated, and evaluated for their cytotoxic activity towards A549, HeLa, and the gastric cancer cells. Through the MTT assay, guavinoside C was shown to have good inhibitory activity against the A549, HeLa, and SGC-7901 cells with IC_50_ values of 4.277, 7.288, and 3.246 μg/mL, respectively. Guavinoside F also has good inhibitory activity with IC_50_ values of 3.729, 7.011, and 3.513 μg/mL against the SGC-7901, A549, and HeLa cells, respectively [21]. Quercetin (Figure 5), a flavonoid commonly found in many vegetables and fruits, has good inhibition against the SGC-7901 and HeLa cells with IC_50_ values of 7.878 and 10 8.260 μg/mL, respectively [21]. Several studies have reported that quercetin inhibits cell proliferation and induces apoptosis in numerous cancer cell lines [58,59,60]. The compounds isolated from the extract also exhibited strong antioxidant activities in the 2,2-diphenyl-1-picrylhydrazyl (DPPH), 2,2’-azino-bis(3-ethylbenzothiazoline-6-sulfonic acid) (ABTS), and ferric reducing antioxidant potential (FRAP) assays [21], which may contribute to their anticancer activity.

The extracts of 27 indigenous Palestinian plants used in traditional medicine were screened for their potential anti-inflammatory and cytotoxic activity. Among the plants, the dichloromethane and methanol (1:1, *v/v*) extract of *P. guajava* L. leaves were found to have the best inhibitory activity on the MCF-7 breast cancer cells with an IC_50_ value of 55 μg/mL at 24 h. Upon pretreatment of the guava leaf extract on L929sA fibrosarcoma cells, a clear inhibition of the nuclear factor kappa-light-chain-enhancer of activated B cells (NF-κB) reporter gene expression was observed [22]. Abnormal activation of the NF-κB has been implicated in the pathogenesis of cancer through its mediation of genes that regulate tumor cell proliferation, survival, and angiogenesis [61]. Further investigation indicates that the repression of NF-κB by the guava extract could be at the NF-κB transactivation level, as pretreatment with the extract did not affect NF-κB/DNA binding [22]. In another investigation, an aqueous *P. guajava* leaf extract was found to exhibit significant inhibitory activity against MCF-7 cell viability at 100 µg, showing the extract’s potential in the treatment of breast cancer [23].

A novel terpenoid saponin glycoside isolated from a fraction of a methanolic guava leaf extract was assessed for its anticancer activity through the MTT assay. At 400 μg/mL, the compound exhibited an average of 99.64% anticancer activity towards MCF-7 cells with an IC_50_ value of 81.50 μg/mL. The compound was also found to exhibit more than 60% apoptotic activity against breast cancer cells [24].

In a study by Sulain et al., the extracts of *P. guajava* leaves were prepared using petroleum ether, methanol, and water as the solvents. The effects of the extracts were evaluated on three cancer cell lines—the HeLa cervical cancer, the MDA-MB-231 breast cancer, and the MG-63 bone osteosarcoma cell lines. While all the *P. guajava* leaf extracts showed no antiproliferative activity against HeLa cells, the petroleum ether extract is the most effective against MDA-MB-231 cells with an IC_50_ value of 4.23 µg/mL followed by the methanol and water extracts at 18.60 and 55.69 µg/mL, respectively. The petroleum ether extract also showed the most inhibitory activity on the cell viability of MG-63 cells, followed by the methanol and water extracts with IC_50_ values of 5.42, 23.25, and 61.88 µg/mL, respectively. However, it should be noted that the anticancer effects of the petroleum ether and methanol extracts were accompanied by cytotoxic effects against the nonmalignant cell Madine Darby canine kidney (MDCK) cells [25].

Kawakami et al. demonstrated that a polyphenol-rich fraction of a *P. guajava* leaf 50% ethanol (*v/v*) extract has growth inhibitory activity on the COLO320DM human colon adenocarcinoma cell line [9], which did not have enzymes that metabolize arachidonic acid [62] and overexpressed prostaglandin H synthase (PGHS) isoforms. PGHS is the rate-limiting enzyme in the synthesis of prostaglandins, the bioactive lipids, which plays vital roles in cancer cell adhesion, migration, and invasion [63]. The overexpression of PGHS-1 and -2 increases the DNA synthesis rate in human colon carcinoma cells. The guava leaf extract was shown to inhibit the PGHS-1 and -2 production in the cancer cells, inhibit the PGHS synthesis, and suppress their cyclooxygenase reaction and DNA synthesis rate [9]. Another study was conducted using the HCT116 colon cancer, Caco-2 epithelial colorectal adenocarcinoma, SW620 lymph node-metastasis colon adenocarcinoma cell lines, and COLO320DM cells to evaluate the anticancer potential of essential oil of *P. guajava* leaf extract from Ecuador. The cell viability of the cancer cells after treatment with the extract was determined using the trypan blue exclusion test. The IC_50_ value of the extract was found to be around 50 μg/mL when tested on the HCT116 cells. At 50 μg/mL, the growth of all three cell lines was reduced by 50-60% after incubation with the extract for 24-48 h. As the extract did not affect the growth of peripheral blood mononuclear cells (PBMCs), it was concluded that the extract’s inhibition of cancer cell growth is due to its antiproliferative rather than the cytotoxic effect [26].

The compounds with anticancer potential were isolated from a 70% ethanol extract of *P. guajava* leaves and characterized through the spectroscopic methods of time-of-flight mass spectrometry (TOFMS), proton nuclear magnetic resonance (^1^H NMR), carbon-13 nuclear magnetic resonance (^13^C NMR), heteronuclear single quantum coherence (HSQC), and heteronuclear multiple bond correlation (HMBC). One new compound, 3,5-dihydroxy-2,4-dimethyl-1-O-(6′-O-galloyl-β-D-glucopyranosyl)-benzophenone (Figure 4), was identified from the extract, as along with the two known compounds, guavinoside B and E. The newly-identified benzophenone and guavinoside B were found to inhibit the HCT116 and HT-29 colon cancer cells in a concentration-dependent manner based on the MTT cell viability assay. The benzophenone showed a more potent inhibitory activity against cell viability than guavinoside B and a stronger activity in inducing cancer cell apoptosis. At 100 μM, the benzophenone inhibited 81.4% of HCT116 cell growth after the cells were treated for 72 h with an IC_50_ value of 60 μM. Guavinoside B inhibited 66.2% of HCT116 cell growth at 100 μM with an IC_50_ value of 80.3 μM. As all three compounds did not show any significant inhibitory effect on the growth of normal CCD-18Co colon cells, the compounds are concluded not to be toxic to normal colon cells [19].

A study on the anticancer effects of aqueous, ethanol, and n-hexane guava leaf extracts was conducted using the MTT assay on HCT116 cells. The aqueous and ethanol extracts were shown to have good inhibitory activity against the colorectal cancer cells with IC_50_ values of 149.66 ± 11.55 and 138.86 ± 12.92 µg/mL, respectively. Among the three extracts, the ethanol extract was also found to have potent antioxidant activities, which may have contributed to its anticancer activity against the HCT116 cells. In addition, the aqueous and ethanol extracts were shown to have inhibition against vascular angiogenesis and impede the branching of the vasculature that supports the growth of tumors [27].

An evaluation of the antioxidant, antibacterial, and antitumoral activities of an ethanolic extract of the *P. guajava* leaves was conducted by Braga et al. Using the MTT assay, the anticancer activity of the extract was evaluated on HeLa and RKO-AS45-1 colorectal cancer cells, resulting in IC_50_ values of 15.6 ± 0.8 and 21.2 ± 1.1 µg/mL, respectively. While the extract showed significant anticancer activity on the HeLa and RKO-AS45-1 cells, it is not cytotoxic towards the Wi-26VA4 lung fibroblast cells [28]. As the extract only showed cytotoxic activity against the cancer cells and not towards the normal cells, the extract seems to exhibit potential as a targeted treatment against these types of cancer.

The DU145 prostate cancer cell line was shown to be inhibited by an aqueous extract of *P. guajava* L. budding leaves in a dose- and time-dependent manner. The viability of the cancer cells was observed to be reduced by the extract at 1 mg/mL and the colony-forming capacity was lowered after the cells were treated with the extract. In addition, cell cycle arrest at G_0_/G_1_ phase has also been observed [29]. In 2009, Chen et al. reported that the budding guava leaf extract contains high concentrations of soluble polyphenolics; one of them is rhamnoallosan, a unique novel peptidoglycan that possesses strong anti-DU145 bioactivity complementary to the other soluble polyphenolics in the extract. At 1 mg/mL, the extract caused a 15.9% reduction in cell viability [30]. The extract has also been shown to inhibit the proliferation of the LNCaP androgen-sensitive human prostate adenocarcinoma in both the absence and presence of androgen. It downregulates the expression of the androgen receptor and prostate-specific antigen of the cells, as well as causes apoptosis and cell cycle arrest at the G_0_/G_1_ phase in a concentration-dependent manner after 48 h in DU145 cells. The extract was also shown to significantly diminish tumor size and prostate-specific antigen serum levels at 1.5 mg/mouse/day in a xenograft mouse tumor model [32].

In a similar study to Chen et al., another aqueous extract of *P. guajava* budding leaves was investigated for its antiangiogenic and antimetastatic effect on DU145 cells. The extract was found to suppress DU145 cell viability in a dose-responsive manner with an IC_50_ value of around 0.57 mg/mL. In addition, the extract was shown to inhibit DU145 cell migration in a dose-responsive manner. The cancer cells treated with the extract also promoted the downregulation of interleukin-6 (IL-6) and interleukin-8 (IL-8), the inhibition of matrix metalloproteinase (MMP)-2 and MMP-9, and the activation of the tissue inhibitor of metalloproteinase (TIMP), all of which contribute to the extract’s inhibitory activity against cancer angiogenesis [31]. These studies contribute towards the validation that the aqueous extracts of *P. guajava* budding leaves possess a strong inhibitory effect against prostate cancer.

The anticancer activity of a methanolic extract of guava leaves was studied using another prostate cancer cell line, PC-3. The hexane fraction from the extract was found to be capable of inducing cytotoxic and apoptotic effects in the cancer cells, arresting the cell cycle at the sub-G_1_ phase, and inducing cell death even at 50 µg/mL [33]. The underlying signaling involves suppression of the AKT/mammalian target of rapamycin (mTOR)/ribosomal p70 S6 kinase (S6K1) and mitogen-activated protein kinase (MAPK) activation pathways. The upregulation of the AKT signaling through the mTOR pathway is observed in various carcinoma cell lines [64,65], which makes it an attractive target for cancer therapeutics. AKT disables apoptosis in cancer cells, promotes tumor metastasis, and contributes towards clinical drug resistance, all of which aids in the survival of the cancer cells [66,67]. The MAPK pathway plays a crucial role in regulating gene expression, cellular growth, and survival [68], so the abnormal signaling of this pathway is important in oncogenesis as it leads to increased or uncontrolled cell proliferation and resistance to apoptosis [69]. In addition, the molecular mechanisms behind the apoptotic potential of the methanolic guava leaf extract correlated with the downregulation of cyclin D_1_, COX-2, and VEGF proteins, which are associated with cell proliferation, cell survival, metastasis, and angiogenesis. All this downstream signaling can potentially contribute to the extract’s inhibitory activity towards PC-3 cell viability [33].

The apoptotic induction activity of an 80% aqueous methanolic extract of *P. guajava* leaves was evaluated on HepG2 human liver carcinoma cells. The cytotoxicity of the extract towards the liver carcinoma cells was evaluated through the MTT assay, in which the extract exhibited a dose-dependent inhibitory activity towards the HepG2 cell growth with cell viability of 81.85%, 70.65%, 53.19%, and 31.09% after exposure of the extract to 5, 20, 50, and 100 μg/mL, respectively. The extract was also shown to induce apoptosis in the cells causing a change of morphology with the cells exhibiting nuclear condensation and DNA fragmentation. The extract also contributed to mitochondrial dysfunction, significantly increasing the reactive oxygen species (ROS) generation of the cells at 20, 50, and 100 μg/mL, and disrupting their mitochondrial membrane potential [34].

A preliminary study of the phytochemicals and the cytotoxic potential of a methanolic extract of *P. guajava* leaves on HeLa cells was investigated by Joseph and Priya. Utilizing the thymidine incorporation assay and comparing the cell count of lysed cells before and after incubation with the extract, *P. guajava* leaf methanolic extract was reported to possess cytotoxic activity towards the cervical cancer cells [35]. Joseph et al. also investigated the cytotoxic activity, along with the antibacterial and antifungal activities of an essential oil extract of *P. guajava* leaves using the thymidine fluorescence assay. The essential oil mixed thoroughly with hexane through the use of micelles exhibited a dose-dependent inhibition of HeLa cell growth. The cells treated with the essential oil showed more prominent growth inhibition and cell shrinkage compared to the control. It is speculated by the authors that the way essential oil exerts its cytotoxicity towards HeLa cells may have been through apoptosis [36].

The antiproliferative activity of the aqueous extract of *P. guajava* leaves was investigated on HeLa and KB human nasopharyngeal epidermoid carcinoma cells by Fathilah et al. With the neutral red cytotoxicity assay, the IC_50_ values of the aqueous *P. guajava* leaf extract on KB and HeLa cells were determined to be 29.0 ±0.4 and 51.0 ± 0.6 µg/mL, respectively [37]. The *P. guajava* leaf extract exhibited potent antiproliferative activity towards the cells, indicating its potential against cervical and oral epidermal cancer.

The cytotoxic effect of a hexane fraction of the methanolic extract of *P. guajava* leaves obtained from Jamaica was investigated on the Kasumi-1 acute myeloblastic leukemia cell line. With an IC_50_ value of 200 ug/mL, the viability of the leukemia cells was significantly inhibited after treatment with the hexane fraction. The cytotoxic activity of the extract was attributed to the presence of gallic acid and flavonoids, such as quercetin and kaempferol (Figure 5) [38]. Interestingly, kaempferol is found in various plant parts, with epidemiological studies indicating that a high intake of kaempferol is associated with decreased incidences of different types of cancer [70]. The anticancer activities of kaempferol include apoptosis, cell cycle arrest, and the inhibition of metastasis and angiogenesis of various cancer cell types [71,72,73].

In a study involving the essential oil extracts of 17 Thai medicinal plants, the antiproliferative effect of the essential oil extract of *P. guajava* L. leaves was investigated on KB and P388 cells using the MTT assay. The guava leaf essential oil showed the most potent antiproliferative activity against KB cells among the 17 medicinal plants with an IC_50_ value of 37.9 µg/mL. Meanwhile, its inhibitory effect on P388 cells was at IC_50_ = 45.4 µg/mL, which is just second to the sweet basil leaf (*Ocimum basilicum*) essential oil extract (IC_50_ = 36.2 µg/mL) [10].

An assessment of the antioxidant, anticancer, and cytotoxic potential of the methanol, chloroform, and hexane extracts of *P. guajava* leaves was conducted by Ashraf et al. Using the MTT assay, the extracts were shown to induce a dose-dependent inhibitory activity against three cancer cell lines: the KBM5 chronic myeloid leukemia, SCC4, oral squamous carcinoma, and U266 multiple myeloma cell lines. Among the extracts tested, the hexane extract exhibited the strongest inhibitory activity against cell viability, resulting in IC_50_ values of 22.73, 22.82, and 20.97 μg/mL against the KBM5, SCC4, and U266 cells, respectively. The chloroform extract also exhibited strong inhibitory activity against U266 cell growth (IC_50_ = 26.72 μg/mL). In addition, the hexane extract suppressed the NF-κB activation induced by TNF-α, which might contribute towards its anticancer potential [39].

A pentacyclic triterpene glycoside, betulinic acid (Figure 4), extracted from guava leaves was evaluated for its anticancer activity against human cholangiocarcinoma (HuCCA) cells. Betulinic acid from the 70% ethanolic guava leaf extract was found to significantly reduce the viability of HuCCA cells in a dose-dependent manner after 24 h with an IC_50_ value of 92.45 µg/mL as assessed by the MTT assay; its anticancer effect was comparable to 5-fluorouracil, a chemotherapy agent for cancer treatment. It was also discovered through Hoechst staining and quantitative real-time PCR that betulinic acid induces the apoptotic signaling pathway of the cancer cells, leading to morphological changes, such as nuclear chromatin condensation and fragmentation [40].

### 2.2. Anticancer Effects of Psidium guajava Fruit Extracts

The cytotoxic activities of the compounds isolated from a petroleum ether extract of mature *P. guajava* fruits from the Yunnan Province, China, were evaluated against five human cancer cell lines—the A549, HCT116, DU145, CCRF-CEM leukemia, and Huh7 hepatocarcinoma cells. The fractionation of the extract and analysis by liquid chromatography-ultraviolet (LC-UV) elucidated 16 meroterpenoids. The meroterpenoid guajadial D was shown to have significant cytotoxic activity against HCT116 with an IC_50_ value of 0.61 ± 0.1 μM based on the MTT assay. Guajadial D had strong cytotoxic activity against the CCRF-CEM cells with an IC_50_ value of 0.87 ± 0.5 μM. 4,5-Diepipsidial A, guadial A, and psiguadial D (Figure 4) had good cytotoxic activities against DU145 having IC_50_ values of 4.79 ± 2.7, 5.35 ± 0.7, 6.08 ± 3.9 μM, respectively. Meanwhile, with IC_50_ values of 2.82 ± 0.6 and 2.93 ± 0.5 μM, 4,5-diepipsidial A and guajadial B were shown to have good cytotoxic activities against the Huh7 cells, respectively, in addition to displaying significant cytotoxicity against A549 cells with IC_50_ values of 0.16 ± 0.03 and 0.15 ± 0.05 μM, respectively [17]. As seen in this and Gao et al.’s study, guajadial D has anticancer potential with cytotoxic activity against various types of cancer cell lines [17,18].

An ethanolic extract from the red guava (*Psidium guajava L*.) fruit, originating from Brazil, was evaluated for its lycopene (Figure 5) content and anticancer activity. Using the MTT assay, this ethanolic extract was found to be cytotoxic towards MCF-7 breast cancer cells, affecting the viability of the cells in a dose-dependent manner and resulting in IC_50_ values of 29.85 and 5.964 µg/mL at 24 and 72 h after incubation with the extract, respectively. Treatment of the MCF-7 cells with the extract resulted in a significant increase in DNA fragmentation and cell cycle arrest at the G_1_ phase, in addition to morphological and mitochondrial membrane potential alterations of the cells [42]. Lycopene is a carotenoid that is a strong antioxidant with known anticancer properties [74,75]. Therefore, the extract’s rich lycopene content, as well as the selectivity of the extract towards breast cancer cells, conveys the extract’s potential as an anticancer treatment [41]. Further studies were conducted with the lycopene-rich extract loaded into lipid-core nanocapsules to stabilize the lycopene against degradation. The loaded nanocapsules were found to significantly reduce the viability of the MCF-7 cells after 24 and 72 h of exposure by 61.47% and 55.96%, respectively, as determined by the MTT assay. The nanocapsules also enhanced the extract’s anticancer activity against the MCF-7 cells, from a cell viability reduction of 24.35% at 6.25 μg/mL to 61.47% after exposure for 24 h [42]. In a later study, the lycopene was loaded into a self-emulsifying drug delivery system (SEDDS) and evaluated for its anticancer activity against DU145 cells. At 3.125 µg/mL, the SEDDS-loaded lycopene significantly reduced DU145 cell viability after 6 h of exposure and had a significantly lower IC_50_ value compared to the free lycopene. Moreover, the histopathological observation of the DU145 cells revealed that the SEDDS-loaded lycopene induces necrotic death of the DU145 cells via a cell death pathway independent of classical necrosis and apoptosis [43].

The anticancer and chemopreventive potential of a guava pulp extract was assessed on breast cancer by conducting an MTT assay on the breast cancer cell lines MCF-7, MDA-MB-231, and MDA-MB-453. The extract showed strong antiproliferative activity against MCF-7 cells in a concentration- and time-dependent manner (IC_50_ = 820 ± 1.22 µg/mL, 680 ± 1.34 µg/mL, and 600 ± 1.03 µg/mL after 24, 48 and 72 h, respectively). The extract’s antiproliferative activity against MDA-MB-231 and MDA-MB-453 is significantly weaker, suggesting that the extract is more effective against estrogen-positive breast cancer cells. In addition, the extract inhibited the cell migration and angiogenesis of MCF-7 cells and significantly increased the apoptotic rate of the cancer cells in vitro [44].

Chadarat et al. conducted a study regarding the cytotoxic effects of the ethanolic extracts of 13 common Thai tropical fruits on leukemic cell lines. The 95% ethanol extracts of the guava leaf, stem, and raw fruit were investigated on K-562, HL-60, U937 myeloid leukemia, and the MOLT-4 acute lymphoblastic leukemia cell lines. The ethanolic extract of the raw guava fruit was found to inhibit the growth of the HL-60 cells with an IC_50_ value of 31.8 ± 2.5 μg/mL. In addition, the extract exhibited no cytotoxic activity towards the normal PBMC line, showing its potential as a treatment for leukemia [45].

The acetone fruit extracts of the IAC-4 variety red guava from São Paulo, Brazil, were investigated by Bontempo et al. Four different acetone extracts were produced using the whole fruit, peel, pulp, and seed. The extracts exhibited antitumoral activity in vitro and ex vivo through the induction of apoptosis and differentiation. The total fruit extract exerted a dose-dependent inhibitory activity on the cell proliferation of NB4 human acute promyelocytic leukemia cells. The extract inhibited the G_1_ cell cycle progression of around 80% of the cells at 1.5 and 3.0 mg/mL. Further investigation of the fruit extracts indicated that the pulp of the red guava fruit is mainly responsible for its apoptotic and cell cycle inhibitory activity, while the peel influences cellular differentiation of the NB4 cells. In addition, the extract exhibited antiproliferative and cell death-inducing effects on MDA-MB 231 cells, suggesting that the anticancer effect of the extract applies to both leukemia and solid tumors [46].

Three biomolecules of interest, apigenin, lycopene, and resveratrol (Figure 5), were isolated from a carotenoid extract of Cuba Enana guava fruit and evaluated for their anticancer potential against LNCaP and human skin melanoma UACC257 cells using the MTT assay and confocal microscopy. Apigenin, a flavonoid commonly found in various fruits and vegetables, was found to have a great inhibitory effect against LNCaP and UACC257 cells, observed through a rapid drop of OD values compared to the untreated cells after 72 h [47]. Apigenin suppresses multiple types of human cancers through various mechanisms, such as inducing cell cycle arrest and apoptosis, inhibiting cell migration and invasion, as well as triggering immune pathways toward the suppression of cancer [76]. Lycopene, a carotenoid that gives fruits and plants their red color, was also found to have a good inhibitory effect against LNCaP, although its effect is slightly less than that of apigenin. In addition, lycopene had a more moderate inhibitory effect on UACC257 cell proliferation compared to its effect on LNCaP [47]. Lycopene was reported to inhibit the growth of different types of cancer cells and prevent chemically induced carcinogenesis in animal models. In addition to its antioxidant properties, lycopene directly inhibited cancer cell proliferation through growth factor regulation, cell cycle arrest and/or apoptosis induction, cell invasion, angiogenesis, and metastasis inhibition [77]. Resveratrol is a biomolecule produced by plants in response to infection and injury. Its inhibitory effect against UACC257 was found to be slightly higher than with lycopene but lower than apigenin, a flavone molecule. However, resveratrol was found to have low antiproliferative activity against LNCaP [47]. Resveratrol targets a number of cellular processes involved in cell apoptosis, cancer angiogenesis and metastasis, as well as increases the chemo-sensitization of cancer cells towards the chemotherapy drugs, 5-fluorouracil and cisplatin [78]. Apigenin and lycopene used in combination caused a sharp decrease in the proliferation of the LNCaP and UACC257 cells after 24 h compared to their effects individually after 72 h. With the apigenin–lycopene combination almost completely inhibiting the growth of the cancer cells, the synergistic effects between the biomolecules in the fruit extract may lead to a better potential cancer treatment [47].

### 2.3. Anticancer Effects of Psidium guajava Seed Extracts

An 80% aqueous acetone extract produced from dry *P. guajava* seeds was investigated by Salib and Michael. Nine phenolic and flavonoid compounds were isolated through fractionalization by column chromatography. The crude extract and two novel phenylethanoid glycosides isolated from it were tested for their anticancer activity on Ehrlich Ascites Carcinoma (EAC) and P388 cells. The crude extract was found to have moderate inhibitory activity against the growth of both the EAC and P388 cells (ED_50_ = 14.6 μg/mL) with the cell concentration determined via counting in a hemocytometer. Meanwhile, the two glycosidic compounds were found to have high inhibitory activity against EAC cells but a low one against P388 cells (ED_50_ = 17.3 and 16.1 μg/mL, respectively) [48].

The in vitro anticancer potential of the seed extracts of edible fruits, pomegranate (*Punica granatum* L.), guava (*P. guajava* L.), and grapes (*Vitis vinifera* L.) was evaluated on A549 cells in a study by Nelson et al. The cytotoxic activity of the ethyl acetate and methanolic seed extracts was analyzed using the MTT assay. While the pomegranate extracts had better activity against the A549 cells, the guava extracts also had good inhibitory activity with IC_50_ values of 60.21 ± 1.35 μg/mL for the ethyl acetate extract and 61.01 ± 1.15 μg/mL for the methanolic extract [49].

A polysaccharide from guava seeds, named Guava seed polysaccharide fraction 3 (GSF3), was found to possess anticancer qualities against MCF-7 cells. This polysaccharide isolated from an aqueous extract of *P. guajava* seeds inhibits the growth of human prostate cancer PC-3 and MCF-7 cells [50,51]. Using the MTT assay, it was found that the direct application of GSF3 significantly inhibited the viability of PC-3 and MCF-7 cells after 24 and 48 h in a dose-dependent manner [51,52]. It was discovered that GSF3′s direct anticancer activity involves promoting cancer cell apoptosis, decreasing the mRNA expression of Bcl-2, a cellular protein that inhibits apoptosis, while increasing the pro-/antiapoptotic mRNA expression ratios in the PC-3 and MCF-7 cells [50,51,52]. Further investigations reveal that GSF3 also indirectly inhibits cancer cell growth through modulating macrophage and splenocyte cytokine secretion profiles in tumor immunotherapy, as well as indirectly contributes to Fas ligand-induced apoptosis in the MCF-7 cells by increasing Fas mRNA expression [51,52].

### 2.4. Anticancer Effects of Psidium guajava Bark and Root Extracts

The acetone and methanolic extracts of 73 medicinal plant species from 44 families, including the bark of *P. guajava* L., were investigated for their anticancer activity against cancer cell lines. The level of inhibition of the extracts on the cancer cell lines was determined using the SRB assay. The acetone extract of the *P. guajava* bark had a strong inhibitory effect on MCF-7 cells with a percentage of inhibition of 83 ± 3% at 200 μg/mL. The acetone extract also had good inhibitory activity against the CAL-27 human epithelial squamous carcinoma, having a percentage of inhibition of 52 ± 17% at 200 μg/mL. The neutral red assay was used to determine the IC_50_ value of the acetone extract of *P. guajava* bark against MCF-7 cells, as well as the cytotoxicity of the extract through the determination of its half-maximum cytotoxicity concentration (CC_50_) value against the Vero monkey kidney epithelial cells. With an IC_50_ value of 115 ± 11 μg/mL, the extract was shown to be effective against breast cancer. However, with a low CC_50_ value of 105 ± 7 μg/mL towards the Vero cells, the extract may not be suitable as an anticancer treatment due to its cytotoxic activity towards normal cells unless further fractionation is undertaken [53].

A set of 18 Cameroonian medicinal plants were investigated for their cytotoxic activity against drug-sensitive and multi-factorial drug-resistant cancer cells. In a preliminary cytotoxicity determination using the resazurin reduction assay, a crude *P. guajava* bark methanolic extract was found to have significant inhibitory activity against the drug-sensitive CCRF-CEM cells with an IC_50_ value of 6.35 ± 1.74 μg/mL. In addition, the extract was tested against the CEM/ADR5000 cells, which is a multidrug-resistant P-glycoprotein-overexpressing subline of the CCRF-CEM cells and the HCT116 (p53+/+) cell line, which resulted in IC_50_ values of 1.29 ± 0.11 and 18.63 ± 1.39 μg/mL, respectively. The extract also has good inhibitory activity on six other cell lines, including the MDA-MB-231-pcDNA3 cells (62.64 ± 0.22 μg/mL) and its resistant subline MDA-MB-231-BCRP clone 23 (27.24 ± 1.96 μg/mL), the HCT116 p53-/- knockout clone (40.63 ± 2.67 μg/mL), the U87MG glioblastoma (28.84 ± 2.19 μg/mL) and its resistant subline U87MG.ΔEGFR (39.86 ± 1.16 μg/mL), and HepG2 (24.63 ± 1.32 μg/mL) cells. The crude *P. guajava* bark methanolic extract was found to alter the cell cycle of the CCRF-CEM cells in a dose-dependent manner, inducing cell cycle arrest at the G_0_/G_1_ phase. At 2 x IC_50_, the extract induced moderate apoptosis of 18.20% through the activation of caspases 3/7 and 9, indicating that its cytotoxic effect might involve an intrinsic mitochondrial pathway. Moreover, the extract also induced the activation of caspase 8, which led to the speculation that the extrinsic apoptotic pathway could also be involved in the signaling. In addition, the extract caused a pronounced depletion of MMP (34.1%) at 2 x IC_50_ and a significant increase in ROS production, indicating that these mechanisms are also involved in the extract’s apoptosis-inducing activity [54].

The anticancer activity of an acetone extract of guava branches was evaluated on the HT-29 cell line. The extract exhibited a dose-dependent inhibition of cell vitality after a 24 h incubation period, with a 30-70% reduction in cell viability compared to the control [55]. The guava branch extract inhibited the colony formation of the cells with an IC_50_ value of less than 100 µg/mL. The extract also caused an increase in the cells’ lactate dehydrogenase (LDH) leakage in a dose-dependent manner, which is indicative of cellular damage [55]. Both the colony formation and the lactate dehydrogenase release assays indicate the extract’s cytotoxicity towards the colon cancer cells. The cytotoxic effect of the extract on HT-29 cells was attributed to its apoptotic activity, as marked chromatin condensation and apoptotic body formation were observed in the cells stained with Hoechst 33342. In addition, the number of apoptotic cells in the sub-G_1_ phase was approximately 30-fold greater in the cells treated with the guava extract compared to the control as assessed by flow cytometry. These results indicate that the acetone extract of the guava branch has great potential as a natural anticancer chemotherapeutic agent for colon cancer through the induction of apoptosis [55].

The ethanolic extracts of *P. guajava* roots, bark, and leaves were shown to exhibit anticancer effects through the immune system [56]. At the early stages of the immune system’s antitumor response, dendritic cells capture and present tumor antigens to initiate the differentiation of naïve T helper (T_h_) cells to produce the cytokines that mediate the antitumor immune response. Naïve T_h_ cells that are activated develop into T_h_1 cells that hamper tumor progression through the production of interleukin (IL)-2 and interferon-gamma (IFN-γ), which increases the phagocytosis of tumor cells and the antitumor immune response [79,80]. T_h_1, together with natural killer (NK) cells, also contributes towards cancer cell apoptosis [81,82] and the inhibition of tumor angiogenesis [83]. The *P. guajava* ethanolic extracts were shown to strongly suppress T regulatory cells, causing a shift in the Th_1_/Th_2_ balance to Th_1_ dominance [56]. The shift towards Th_1_ dominance exerts favorable anticancer immunological effects [84], while Th_2_-mediated immunity is associated with both promoting and inhibiting tumor growth. Mice pretreated with the oral administration of *P. guajava* extracts also showed an impeded growth of inoculated B16 melanoma tumors that are approximately one-seventh of the size of the tumors in the controls after 20 days. These results indicate that the extracts have an antiallergic action against Th_2_ cell-mediated allergy, in addition to having a protective action against tumor development [56].

### 2.5. In Vivo Anticancer Effects of Psidium guajava Extracts

The ethanolic combination of *P. guajava* extract made from the plant’s leaf, bark, and root was also tested by Seo et al. using an in vivo model of B16 skin melanoma cells in mice. As mentioned above, the mice that were pretreated with the extract formed tumors that were approximately one-seventh of the size of the tumors found in control after 20 days. However, the extract did not have an effect on the size of existing tumors, showing that the extract had more of a chemopreventive effect rather than a therapeutic effect on the tumors [56]. The result is in line with the in vitro results regarding the extract’s effects on tumors via the immune system with the extract downregulating T cell activity in the mice and protecting the mice from tumor development.

In addition to the in vitro studies on DU145 and LNCaP cells, the aqueous *P. guajava* budding leaf extract by Chen et al. was also tested using an LNCaP xenograft mouse tumor model. Oral administration of 1.5 mg/mouse/day led to significantly reduced tumor sizes with diminished prostate-specific antigen in the serum. The findings are compatible with the in vitro studies, which found that the extract induces apoptosis and cell cycle arrest in LNCaP cells [32].

The meroterpene-enriched fraction derived from the dichloromethane extract of *P. guajava* leaves, which was found by Rizzo et al. to have significant anticancer activity on multiple cell lines mentioned above, was also tested in vivo using subcutaneous solid EAC (Ehrlich murine breast adenocarcinoma) on adult female mice. The *P. guajava* fraction treatment was administered every 3 days at 10, 30, and 50 mg/kg. After 21 days, it was found that the treatment significantly inhibited tumor growth and was equally effective as the positive controls (doxorubicin and tamoxifen) with less toxicity [8]. An increase in both uteri size and weight of the treated mice suggests that the *P. guajava* treatment has a phytohormonal and estrogen-like effect on uterus proliferation and indicates that it may have a mechanism of action similar to that of tamoxifen, an antihormonal chemotherapeutic medication for estrogen receptor-positive breast and ovarian cancer [8,85]. Further research on an enriched guajadial fraction obtained from the *P. guajava* leaf extract demonstrates that its tumor inhibitory activity may have been through the estrogen receptors, similar to tamoxifen, as guajadial is structurally similar to this breast cancer drug. In addition to the guajadial fraction’s in vitro activity against estradiol-sensitive breast cancer cells, in vivo studies indicated that it inhibits the proliferative effect of estradiol on the uterus of prepubescent rats as well [20].

To further confirm the anticancer potential of a guava fruit pulp extract on estrogen receptor-positive breast cancer, an in vivo study was conducted using N-methyl-N-nitrosourea (MNU)-induction of mammary cancer in female Sprague Dawley rats. Through the measurement of growth rate, feed consumption efficiency, tumor parameters (weight, number of tumors, volume, tumor incidence, and latency period), estrogen and progesterone expressions, and nucleic acid content of the rats, the pulp extract was found to be successful in suppressing carcinogen-induced tumor incidence and multiplicity. A significant decrease in the expression of estrogen and progesterone in the animals treated with both the pulp extract and tamoxifen was also observed compared to the control animals, which supports the inference about the anticancer potential of the extract against estrogen-receptor-positive breast cancers [44].

After the biological assays on EAC and P388 cells, the in vivo effects of the 80% aqueous acetone extract of *P. guajava* seeds were tested by Salib and Michael using intraperitoneal transplantation of EAC cells in female Swiss albino mice. Oral administration of the extract and the two glycosidic compounds isolated from the extract were found to increase the survival time of mice bearing the adenocarcinoma tumor (18 to 24 days) compared to the control (10 days) [48].

## 3. Discussion

Cancer is a common death-causing disease that exerts a heavy societal burden that continues to grow worldwide. Unfortunately, a significantly large portion of patients suffering from cancer globally do not have access to timely diagnosis and quality treatment as the healthcare systems in low- and middle-income countries are the least prepared to manage the cancer burden [7]. *P. guajava* is a commonly found plant in Asia, South America, Africa, and Europe and is highly valued as a food crop. It has a good nutritious value, possessing an array of useful natural compounds, such as phenolics, flavonoids, carotenoids, and triterpenoids. The plant and its fruits are abundantly available in middle- to low-income countries, contributing to its suitability for the development of economical cancer preventative and treatment solutions. The chemopreventives and treatments sourced from local plants would be more affordable as raw materials can be processed locally into the final product and greatly reduce transportation and shipping costs. It would address the issue of patients from low- and middle-income countries who are forced to abandon treatment due to high costs.

The farming and processing of the guava fruit leaves behind the peel, leaves, seeds, and bark of the *P. guajava* tree. Rather than discarding these leftover farming by-products, they can be repurposed as materials for producing chemopreventives and cancer treatments. This could be an extra source of income for the farmers, especially for those from low- and middle-income countries, and contributes towards reducing agricultural waste. Most of the studies had focused on the anticancer effects of *P. guajava* leaves, which is a good source of material for the production of anticancer treatment without cutting into agricultural profits. In addition to the leaves, the bark and roots of the *P. guajava* plant have traditionally been used as a treatment for a wide array of diseases. The leaves were used to treat toothaches and skin wounds in India. In Southeast Asia, the leaves, bark, and fruits were used to treat microbial infections, hypertension, and gastrointestinal issues [86]. The use of local plants as a treatment for diseases is much more culturally accepted in developing countries, which helps boost economic self-reliance by reducing drug import [87].

From this review, we can see that the research on the anticancer potential of *P. guajava* has been more focused on the plant’s leaves. The various biomolecules present in *P. guajava* and the possible mechanisms by which these compounds present in the *P. guajava* plant extracts might contribute to the anticancer properties attributed to this plant and are summarized in Table 2. More research should also be conducted on the chemopreventive potential of the guava fruit, as it is an economical and easily accessible food source. The incorporation of guava fruit into the daily diet of the community would ameliorate some of the issues when it comes to cancer management in developing countries.

In addition to the extracts and their fractions, a number of active bio-compounds identified to be present in the *P. guajava* extracts were shown to contribute to the extracts’ anticancer activity against human cancer cells through different mechanisms, such as their antiproliferative, antioxidant, and angiogenic activities, as well as through their effects on the immune system targeting cancer cells and tumors. Further research on these bioactive molecules is required to elucidate the mechanisms and their roles against human cancers, as well as their synergistic effects within the extracts that would make the extracts to be utilized as a better anticancer solution.

While there have been clinical trials of the use of *P. guajava* extracts as treatments for infectious diarrhea [88,89,90], diabetes [91,92], and pain management [93,94], until now, no clinical trial has been reported investigating the anticancer effect of *P. guajava*. Future clinical trials are necessary to establish the safety and efficacy of *P. guajava* extracts as a treatment for human ailments. Meanwhile, clinical trials regarding the feasibility of *P. guajava* extracts as a possible treatment for cancer, adjuvant therapy, and chemopreventative agent could be further explored.

## 4. Conclusions

*Psidium guajava* extracts using various plant parts were investigated for their effects on different human cancer cell lines and animal models, all of which have showcased the potential anticancer effects of the plant. With a lack of any known toxicity towards human cells, *P. guajava* extracts can be inferred to be a viable candidate in the search for safe and affordable treatments for various types of human cancers in the future. The leaves of the plant, especially, were found to contain various bioactive compounds that potentially have anticancer and chemopreventative activities. Meanwhile, more studies should be conducted investigating the potential anticancer effects of guava fruit. As a commonly eaten fruit in tropical and subtropical regions, a greater understanding of the fruit’s potential as a chemopreventive food will help alleviate the cancer burden.

## Figures and Tables

**Figure 1 life-13-00346-f001:**
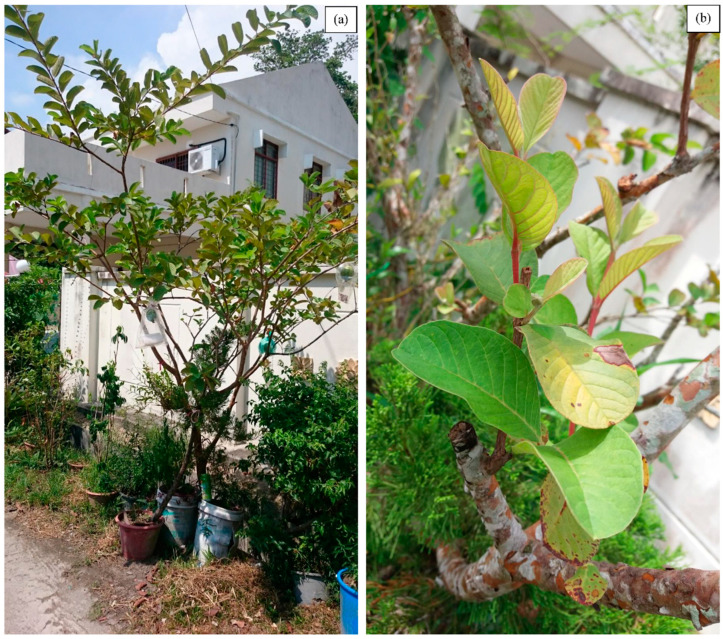
Photographs depicting (**a**) the whole *Psidium guajava* (guava) plant and (**b**) a close-up of the leaves.

**Figure 2 life-13-00346-f002:**
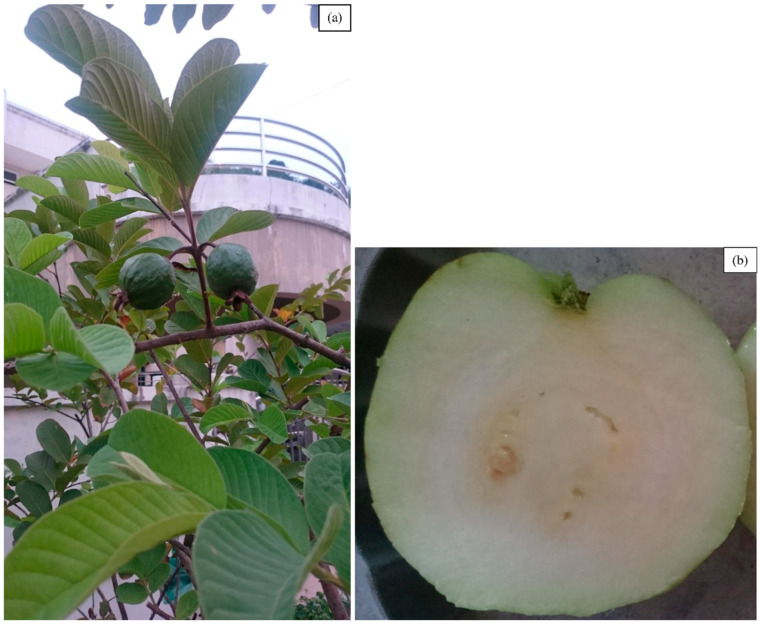
Photographs depicting (**a**) a close-up of unripe *Psidium guajava* (guava) fruits on the plant and (**b**) a cross-section of a ripe guava fruit.

**Figure 3 life-13-00346-f003:**
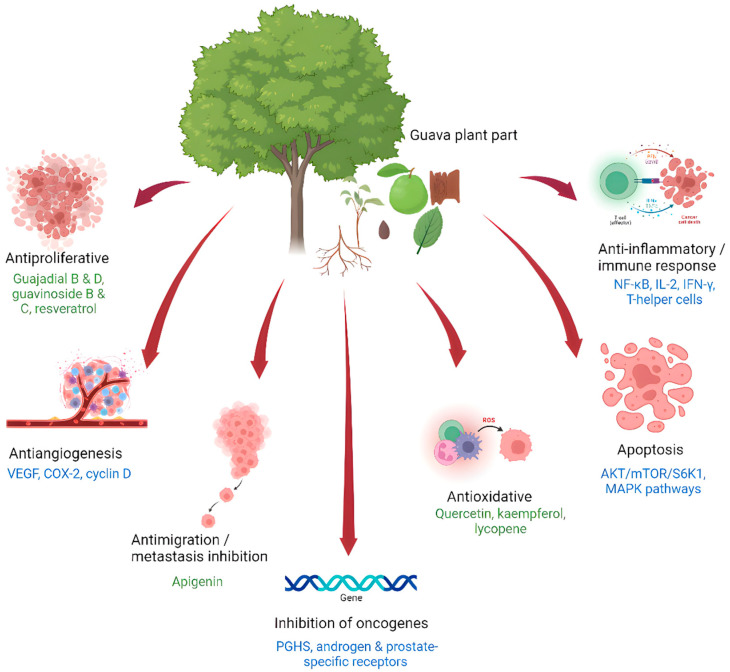
Summary of the various biomolecules and the different mechanisms involved in the anticancer properties of *Psidium guajava* plant extracts. This figure was created using Biorender.com.

**Figure 4 life-13-00346-f004:**
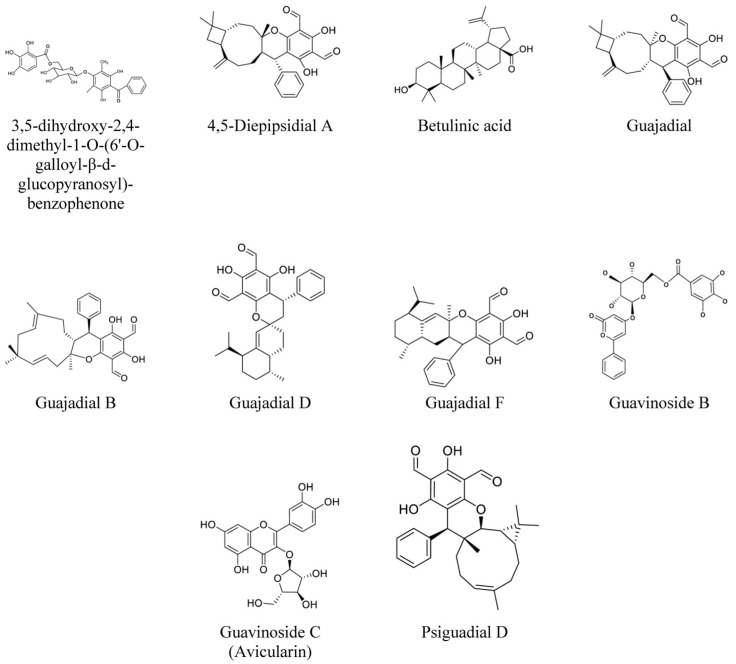
Chemical structures of benzophenone, terpenoids, and glycosides with anticancer properties present in *Psidium guajava* extracts.

**Figure 5 life-13-00346-f005:**
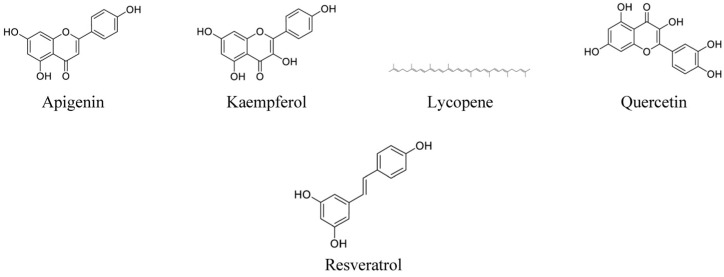
Chemical structures of flavonoids and carotenoids with anticancer properties present in *Psidium guajava* extracts.

**Table 1 life-13-00346-t001:** List of *P. guajava* plant material(s) and the different extracts with in vitro anticancer activity against human cancer cell lines.

Part of Plant	Type of Extracts	Type of Cancer	References
Leaves	Dichloromethane	NCI-H460 (lung cancer), MCF-7 (breast cancer), HT-29 (colon cancer), PC-3 (prostate cancer), K562 (leukemia), 786-0 (kidney cancer), OVCAR-3, NCI/ADR-RES (ovarian cancer), and UACC-62 (melanoma)	Rizzo et al., 2014 [8]
Leaves	Dichloromethane	MCF-7 and MCF-7 BUS (breast cancer)	Bazioli et al., 2020 [20]
Leaves	Methanol	A549 (lung cancer), MCF-7 (breast cancer), SMMC-7721 (cervical cancer), and HL60 (leukemia)	Gao et al., 2013 [18]
Leaves	75% ethanol	A549 (lung cancer), HeLa (cervical cancer), and SGC-7901 (gastric cancer)	Feng et al., 2015 [21]
Leaves	Dichloromethane and methanol (1:1, *v/v*)	MCF-7 (breast cancer)	Kaileh et al., 2007 [22]
Leaves	Aqueous	MCF-7 (breast cancer)	Sukanya et al., 2017 [23]
Leaves	Methanol	MCF-7 (breast cancer)	Manikyam et al., 2021 [24]
Leaves	Petroleum ether, methanol, water	MDA-MB-231 (breast cancer), MG-63 (bone osteosarcoma)	Sulain et al., 2012 [25]
Leaves	Methanol	MDA-MB-231 (breast cancer), MG-63 (bone osteosarcoma)	Sulain et al., 2012 [25]
Leaves	Aqueous	MDA-MB-231 (breast cancer), MG-63 (bone osteosarcoma)	Sulain et al., 2012 [25]
Leaves	Polyphenol-rich fraction 50% ethanol (*v/v*) extract	COLO320DM (colon cancer)	Kawakami et al., 2009 [9]
Leaves	Essential oil	HCT116, Caco-2, and SW620 (colon cancer)	Radice et al., 2017 [26]
Leaves	70% ethanol	HCT116 and HT-29 (colon cancer)	Zhu et al., 2019 [19]
Leaves	Aqueous, Ethanol	HCT116 (colon cancer)	Lok et al., 2020 [27]
Leaves	Ethanol	RKO-AS45-1 (colorectal cancer) and HeLa (cervical cancer)	Braga et al., 2014 [28]
Leaves	Aqueous	DU145 (prostate cancer)	Chen et al., 2007 [29]; Chen et al., 2009 [30]
Leaves	Aqueous	DU145 (prostate cancer)	Peng et al., 2011 [31]
Leaves	Aqueous	LNCaP (prostate cancer)	Chen et al., 2010 [32]
Leaves	Hexane fraction of methanol extract	PC-3 (prostate cancer)	Ryu et al., 2012 [33]
Leaves	80% aqueous methanol	HepG2 (liver cancer)	Nguyen et al., 2019 [34]
Leaves	Methanol	HeLa (cervical cancer)	Joseph and Priya, 2010 [35]
Leaves	Essential oil	HeLa (cervical cancer)	Joseph et al., 2010 [36]
Leaves	Aqueous	HeLa (cervical cancer) and KB (mouth epidermal cancer)	Fathilah et al., 2010 [37]
Leaves	Hexane fraction of methanol extract	Kasumi-1 (leukemia)	Levy and Carley, 2012 [38]
Leaves	Essential oil	P388 (leukemia) and KB (mouth epidermal cancer)	Manosroi et al., 2006 [10]
Leaves	Hexane	KBM5 (leukemia), U266 (multiple myeloma), and SCC4 (tongue carcinoma)	Ashraf et al., 2016 [39]
Leaves	Chloroform	U266 (multiple myeloma)	Ashraf et al., 2016 [39]
Leaves	70% ethanol	HuCCA (cholangiocarcinoma)	Phonarknguen et al., 2022 [40]
Fruits	Petroleum ether	A549 (lung cancer), HCT116 (colon cancer), CCRF-CEM (leukemia), DU145 (prostate cancer), and Huh7 (liver cancer)	Qin et al., 2017 [17]
Fruits	Ethanol	MCF-7 (breast cancer)	dos Santos et al., 2018 [41]
Fruits	Ethanol	MCF-7 (breast cancer)	Vasconcelos et al., 2020 [42]
Fruits	Ethanol	DU145 (prostate cancer)	Vasconcelos et al., 2021 [43]
Fruits, pulp	-	MCF-7, MDA-MB-231, and MDA-MB-453 (breast cancer)	Karia et al., 2019 [44]
Fruits	Ethanol	HL-60 (leukemia)	Chadarat et al., 2010 [45]
Fruits, pulp	Acetone	MDA-MB 231 (breast cancer) and NB4 (leukemia)	Bontempo et al., 2012 [46]
Fruits	Carotenoid	LNCaP (prostate cancer) and UACC257 (melanoma)	Priam et al., 2021 [47]
Seeds	80% Aqueous acetone	EAC (skin cancer) and P388 (leukemia)	Salib and Michael, 2004 [48]
Seeds	Methanol, ethyl acetate	A549 (lung cancer)	Nelson et al., 2019 [49]
Seeds	Aqueous	MCF-7 (breast cancer), PC-3 (prostate cancer)	Lin and Lin, 2016 [50], 2020 [51], 2021 [52]
Barks	Acetone	MCF-7 (breast cancer), CAL 27 (tongue carcinoma)	Cates et al., 2013 [53]
Barks	Methanol	MDA-MB-231 (breast cancer), HCT116 (colon cancer), HepG2 (liver cancer), CCRF-CEM (leukemia), and U87MG (brain cancer)	Mbaveng et al., 2018 [54]
Branches	Acetone	HT-29 (colon cancer)	Lee and Park, 2010 [55]
Leaves, bark, and roots	Ethanol	B16 melanoma (skin cancer)	Seo et al., 2005 [56]

**Table 2 life-13-00346-t002:** Bioactive molecules that are isolated from *P. guajava* plant material(s) and their anticancer mechanisms against human cancer cell lines.

Bioactive Molecules of Interest	Part of Plant	Type of Cancer Cells	Anticancer Mechanism(s)	References
3,5-dihydroxy-2,4-dimethyl-1-O-(6’-O-galloyl-β-d-glucopyranosyl)-benzophenone	Leaves	HCT116 and HT-29 (colon cancer)	Antiproliferative, apoptosis	Zhu et al., 2019 [19]
4,5-Diepipsidial A	Fruits	DU145 (prostate cancer), Huh7 (liver cancer)	Antiproliferative	Qin et al., 2017 [17]
Apigenin	Fruits	LNCaP (prostate cancer) and UACC257 (melanoma)	Antiproliferative, apoptosis, antimigration, and immune response	Priam et al., 2021 [47]
Betulinic acid	Leaves	HuCCA (cholangiocarcinoma)	Antiproliferative and apoptosis	Phonarknguen et al., 2022 [40]
Guajadial	Leaves	A549 and H1650 (lung cancer), MCF-7 and MCF-7 BUS (breast cancer)	Antiproliferative and antimigration	Rizzo et al., 2014 [8]
Guajadial B	Fruits	Huh7 (liver cancer)	Antiproliferative	Qin et al., 2017 [17]
Guajadial D	Leaves, Fruits	A549 (lung cancer), CCRF-CEM and HL60 (leukemia), HCT116 (colon cancer), and SMMC-7721 (cervical cancer)	Antiproliferative	Gao et al., 2013 [18] and Qin et al., 2017 [17]
Guajadial F	Leaves	A549 (lung cancer), MCF-7 (breast cancer), HL60 (leukemia), and SMMC-7721 (cervical cancer)	Antiproliferative	Gao et al., 2013 [18]
Guavinoside B	Leaves	HCT116 and HT-29 (colon cancer)	Antiproliferative	Zhu et al., 2019 [19]
Guavinoside C (Avicularin)	Leaves	A549 (lung cancer), HeLa (cervical cancer), and SGC-7901 (gastric cancer)	Antiproliferative	Feng et al., 2015 [21]
Kaempferol	Leaves	Kasumi-1 (leukemia)	Apoptosis, antiangiogenesis, and antimigration	Levy and Carley, 2012 [38]
Lycopene	Fruits	DU145 (prostate cancer), LNCaP (prostate cancer), MCF-7 (breast cancer), and UACC257 (melanoma)	Antiproliferative, antioxidative, antiangiogenesis, and antimigration	dos Santos et al., 2018 [41], Vasconcelos et al., 2020 [42], Vasconcelos et al., 2021 [43], and Priam et al., 2021 [47]
Psiguadial D	Fruits	DU145 (prostate cancer)	Antiproliferative	Qin et al., 2017 [17]
Quercetin	Leaves	A549 (lung cancer), HeLa (cervical cancer), Kasumi-1 (leukemia), and SGC-7901 (gastric cancer)	Antiproliferative, apoptosis, and antioxidative	Feng et al., 2015 [21] and Levy and Carley, 2012 [38]
Resveratrol	Fruits	UACC257 (melanoma)	Antiproliferative, apoptosis, antiangiogenesis, and antimigration	Priam et al., 2021 [47]

**Table 3 life-13-00346-t003:** List of *P. guajava* plant material(s) and the different extracts with in vivo anticancer activities in animal models.

Part of Plant	Type of Extracts	Type of Cancer	In Vivo Animal Model	References
Leaves, bark, and roots	Ethanol	B16 melanoma (skin cancer)	Subcutaneous inoculation of cancer cells into female B6 and BALB/c mice	Seo et al., 2005 [56]
Leaves	Aqueous	LNCaP (prostate cancer)	Subcutaneous inoculation of cancer cells into male BALB/c mice	Chen et al., 2010 [32]
Leaves	Dichloromethane	EAC (skin cancer)	Subcutaneous inoculation of cancer cells into female BALB/c mice	Rizzo et al., 2014 [8]
Fruits, pulp	-	Breast cancer	MNU-induced rat mammary tumors in female Sprague Dawley rats	Karia et al., 2019 [44]
Seeds	80% Aqueous acetone	EAC (skin cancer)	Intraperitoneal inoculation into female Swiss albino mice	Salib and Michael, 2004 [48]

## Data Availability

No new data were created or analyzed in this study. Data sharing is not applicable to this article.

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
