# Peer review of "The Anticancer Potential of Psidium guajava (Guava) Extracts"

_life, 2023, doi:10.3390/life13020346_

Round 1

Reviewer 1 Report

In this study, the authors evaluated the review of” Anticancer potential of Psidium guajava (Guava) extracts”. The study is relevant and interesting, although, it has lots of similarities with previous published papers. However, the manuscript should be accepted after major corrections.

1.      The author should develop a table that summaries bioactive compounds isolated from Psidium guajava, method of isolation, study models, mechanism of action and types of cancer they inhibit.

2.      The authors should redraw the structures of Psidium guajava bioactive compounds with anticancer activities using good software such as ChemDraw.

3.      The authors should review extensively studies on anticancer activities of bioactive compounds isolated from Psidium guajava.

4.      Mechanism of action of bioactive compounds isolated Psidium guajava and Psidium guajava extracts should be improved.

5.      The authors should rewrite the conclusion of this study; they should include the major findings and relevance of this study as well as future prospects.

6.      English language needs revision (few typographical and grammatical errors were detected).

7.      The authors should use the journal format to compile all the listed references. For instance, references 28, 31 and 26, 27 are formatted differently.

Author Response

Point 1: The author should develop a table that summaries bioactive compounds isolated from Psidium guajava, method of isolation, study models, mechanism of action and types of cancer they inhibit.

Response 1: We thank the reviewer for the constructive suggestion. A new table (Table 2) has now been added in the revised manuscript, which summarizes the types of bioactive compounds that could be isolated from the different plant parts of P. guajava, along with their associated mechanisms contributing to their activities to inhibit the different types of cancer.

Point 2: The authors should redraw the structures of Psidium guajava bioactive compounds with anticancer activities using good software such as ChemDraw.

Response 2: The chemical structures have been redrawn using ChemDraw in the revised manuscript.

Point 3: The authors should review extensively studies on anticancer activities of bioactive compounds isolated from Psidium guajava.

Response 3: As mentioned above for point 1, a new table summarizing the bioactive compounds and their anticancer activities have been added. However, only about one-half of the reported few studies isolated the bioactive compounds from the extracts, while only a few studies have conducted further studies on the anticancer activity of those compounds. Many of the compounds’ anticancer activity have been attributed to their antioxidative potential and their effects on the immune system, which are derived from the studies of these bioactive compounds from other sources which are unrelated to P. guajava; therefore, elaborating too much on the bioactive compounds without direct supporting studies derived from P. guajava would be a vague presentation for a review article.

Point 4: Mechanism of action of bioactive compounds isolated Psidium guajava and Psidium guajava extracts should be improved.

Response 4: Even though there has been a number of bioactive compounds isolated from P. guajava and its extracts, very little studies have investigated their mechanisms of action in terms of anticancer activity, which we have already described in the manuscript and also have listed in Table 2 of the revised manuscript.

Point 5: The authors should rewrite the conclusion of this study; they should include the major findings and relevance of this study as well as future prospects.

Response 5: The conclusion has been rewritten as per the reviwer’s suggestions.

Point 6: English language needs revision (few typographical and grammatical errors were detected).

Response 6: The typographical and grammatical errors have been corrected.

Point 7: The authors should use the journal format to compile all the listed references. For instance, references 28, 31 and 26, 27 are formatted differently.

Response 7: References 26, 27, and 28 are journal papers while reference 31 is a conference proceeding, hence the difference in reference formatting. Nevertheless, the references have now been edited to fit MDPI’s journal format.

Reviewer 2 Report

The manuscript is original and interesting, but it should be revised it by an English expert.

The authors should add a paragraph to highlight the main analytical methods and techniques used for the identification and quantification of bioactive compounds in this plant materials.

Moreover, a table with the indication of bioactive compounds and their range amounts should be added.

The conclusion should be better linked to the study aims.

Author Response

Point 1: The manuscript is original and interesting, but it should be revised it by an English expert.

Response 1: The new version of the manuscript has been revised for English language.

Point 2: The authors should add a paragraph to highlight the main analytical methods and techniques used for the identification and quantification of bioactive compounds in this plant materials.

Response 2: As per the reviewer’s suggestion, a paragraph has been added to highlight the analytical techniques for the isolation and identification of bioactive compounds in the plant extracts.

Point 3: Moreover, a table with the indication of bioactive compounds and their range amounts should be added.

Response 3: A table (Table 2) has been added to summarize the types of bioactive compounds that could be isolated from the different plant parts of P. guajava, and their associated mechanisms they utilize to inhibit the different types of cancer. However, their range amounts of the particular bioactive compounds that can be isolated are difficult to include as not all the articles provide the data, only stating the presence of those molecules.

Point 4: The conclusion should be better linked to the study aims.

Response 4: The conclusion has been linked to connect the aim of the study as per the reviewer’s suggestion.

Reviewer 3 Report

The authors studied thoroughly the anticancer potential of Guava (Psidium guajava) extract reviewing 89 scientific papers.

The manuscript is suitable for publication after minor corrections.

Line 94.: then the amount…

The number of the corresponding references are missing from the last column of Table 1. and Table 2.

The explanation of the abbreviations should be either in the list of abbreviations or in the text, when it occurs at first, e.g. line 110: OD, line 473: UACC.

Lines 119-126: there is no reference in this paragraph.

Lines 139-166: this paragraph is too long. Line 158. should start as a new paragraph summarizing references [17] and [18]. Guajadial…

Line 162: guava leaf extract by Rizzo et al…. line 166: [18], but this paper was written by Bazioli et al.

Line 244: 3,5-dihydroxy-2,4-dimethyl-1-O-(6’-O-galloyl-β-D-glucopyranosyl)-benzophenone is mentioned. The structure of this compound is missing from this paragraph.

Lines 435-442: The study by Antasaveta et al. is detailed here, but ref [65] is given at the end of the paragraph, which was published by Chadarat et al.

Figure 3 is unreadable, sometimes incorrect, because C-C bonds are missing from the structures of aromatic rings. Two separated figures one on guajadial A B, D, F, dipepsidial A, and psiguadial D, etc., another one for the flavonoid derivatives: apigenin, kaempferol, quercetin, resveratrol would be more useful after the corresponding paragraph, where they were mentioned, e.g. line 402, line 482 respectively. The structure of lycopene would be informative after line 425.

The number of the Figures should be always mentioned in the text.

A better place would be for the structure of betulinic acid after line 382, because this paragraph is focusing on this compound.

The list of references needs a careful control:

Everywhere, the abbreviated first name of the authors is coming after the family name.

[20] Bioorganic and Medicinal Chemistry Letters

[65] A space is missing.

[67] Preprint (Version 1) The bold character is unnecessary.

Author Response

Point 1: Line 94.: then the amount…

Response 1: It has been corrected in the revised manuscript.

Point 2: The number of the corresponding references are missing from the last column of Table 1. and Table 2.

Response 2: The numbers for the references have been added to Tables 1, 2, and 3.

Point 3: The explanation of the abbreviations should be either in the list of abbreviations or in the text, when it occurs at first, e.g. line 110: OD, line 473: UACC.

Response 4: The full expansion of OD has been added to the list of abbreviations. The cell line name for UACC at line 473 has been clarified.

Point 5: Lines 119-126: there is no reference in this paragraph.

Response 5: This paragraph consists of general statements regarding the contents in the main body of the manuscript.

Point 6: Lines 139-166: this paragraph is too long. Line 158. should start as a new paragraph summarizing references [17] and [18]. Guajadial…

Response 6: The studies done in references [17] and [18] builds upon the study in reference [8], as they utilized a guajadial fraction obtained from the crude dichloromethane guava leaf extract. Therefore, it is more suitable for these studies to be grouped into a single paragraph, which appears to be a long paragraph.

Point 7: Line 162: guava leaf extract by Rizzo et al…. line 166: [18], but this paper was written by Bazioli et al.

Response 7: As mentioned in the answer for the previous response, the study done in reference [18] expands upon the work by Rizzo et. al, using the guava leaf extract that Rizzo et. al. had originally produced to create a guajadial fraction used for their studies. Therefore, the statement is mentioned as a guava leaf extract produced by Rizzo et. al.

Point 8: Line 244: 3,5-dihydroxy-2,4-dimethyl-1-O-(6’-O-galloyl-β-D-glucopyranosyl)-benzophenone is mentioned. The structure of this compound is missing from this paragraph.

Response 8: The chemical structure has now been added, along with other chemical structures and not among the paragraph in order to not clutter the manuscript with figures of chemical structures in between the text.

Point 9: Lines 435-442: The study by Antasaveta et al. is detailed here, but ref [65] is given at the end of the paragraph, which was published by Chadarat et al.

Response 9: Thank you for your comment. The author’s first name was mistakenly quoted instead of their last name. It has now been corrected in the revised version.

Point 10: Figure 3 is unreadable, sometimes incorrect, because C-C bonds are missing from the structures of aromatic rings. Two separated figures one on guajadial A B, D, F, dipepsidial A, and psiguadial D, etc., another one for the flavonoid derivatives: apigenin, kaempferol, quercetin, resveratrol would be more useful after the corresponding paragraph, where they were mentioned, e.g. line 402, line 482 respectively. The structure of lycopene would be informative after line 425.

Response 10: The figures of the chemical structures have been edited. The structures were separated into two figures (Figures 2 and 3) as per the reviewer’s suggestion and have been moved to their corresponding paragraphs in the revised manuscript.

Point 11: The number of the Figures should be always mentioned in the text.

Response 11: The numbers of the figures have been included appropriately now.

Point 12: A better place would be for the structure of betulinic acid after line 382, because this paragraph is focusing on this compound.

Response 12: The chemical structures of the compounds are grouped together so that the manuscript looks more streamlined and not crowded with figures of individual chemical structures in between the paragraphs of the text.

Point 13: The list of references needs a careful control:

Everywhere, the abbreviated first name of the authors is coming after the family name.

[20] Bioorganic and Medicinal Chemistry Letters

[65] A space is missing.

[67] Preprint (Version 1) The bold character is unnecessary.

Response 13: The corrections to the references have been made.

Reviewer 4 Report

Title: The anticancer potential of Psidium guajava (Guava) extracts

In general, this review paper contains abundance of information on the anticancer properties of several Guava parts. I do have some minor comments.

1.       Please provide a high resolution for Figure 3.

2.       Guava extract safety evaluation must be added. This is a very crucial subject to discuss. Selectivity index must also be included for the in vitro studies.

3.       Do guava extract clinical trials exist? If not, discuss about this limitation.

4.       Because this is a review article, 70 % of the references must be recent—no older than 2010.

Author Response

Point 1: Please provide a high resolution for Figure 3.

Response 1: Figure 3 has been separated into 2 new figures (Figure 2 and 3) with high resolution.

Point 2: Guava extract safety evaluation must be added. This is a very crucial subject to discuss. Selectivity index must also be included for the in vitro studies.

Response 2: For most of the in vitro studies highlighted in this manuscript, a control study using a normal cell type is used to assess the safety of the extracts on non-cancerous cells. These are discussed together with the studies with their corresponding IC50 or EC50 values compared to those of the cancer cells, wherever possible.

Point 3: Do guava extract clinical trials exist? If not, discuss about this limitation.

Response 3: Clinical trials for guava extracts exist, but not for its anticancer potential. Discussions about this limitation has been added.

Point 4: Because this is a review article, 70 % of the references must be recent—no older than 2010.

Response 4: We agree with the reviewer’s suggestion; however, there are multiple studies before 2010 that are vital in the research of showcasing P. guajava’s potential as a treatment for cancer. Therefore, they are also cited in this manuscript. In addition, the recent elaborated investigations are based on the various anticancer studies performed earlier, so we have included important old studies in order to give due credits to those significant foundational studies.

Reviewer 5 Report

The anticancer potential of Psidium guajava (Guava) extracts

This work explores the potential of different parts of the guava plant, as well as the molecules that can be extracted from them, in terms of the treatment of human cancers.

Anticancer Effects of Psidium guajava Leaf Extracts

A table would support this section, or else a schematic or figure.

Figure 3. It is of low quality. I suggest using a program to draw structures. Similarly, I suggest IC50 or ED50 be put on each of the compounds related to free radical activity or cytotoxicity.

Supplementary material

Make a general table of each of the parts of the plants, and give the IC50 values of extracts.

Make a table with the IC50 values of the compounds isolated from the plant.

Figure 4. Mention the compounds that correspond to each activity, if they do not exist in the literature, place the extracts either from leaves, bark, roots, flowers or fruits.

 Conclusion

Based on the review carried out, mention which part of the plant has the greatest anticancer activity, and order them in order of activity.

Author Response

Point 1: Anticancer Effects of Psidium guajava Leaf Extracts

A table would support this section, or else a schematic or figure.

Response 1: The information in this section is summarised in Table 1, in which they are grouped based on the plant part the extracts originated from.

Point 2: Figure 3. It is of low quality. I suggest using a program to draw structures. Similarly, I suggest IC50 or ED50 be put on each of the compounds related to free radical activity or cytotoxicity.

Response 2: As per the reviewer’s suggestion, the chemical structures from Figure 3 have been redrawn, and separated into 2 figures (Figures 2 and 3) with a higher quality than the previous version. Unfortunately, it is difficult to include the IC50 or ED50 of each of the compounds as most of the studies have investigated the anticancer activity of the extracts as a whole (without precise IC50 concentrations) and not as individual pure compounds to precisely define their effective half maximal effects.

Point 3: Supplementary material

Make a general table of each of the parts of the plants, and give the IC50 values of extracts.

Make a table with the IC50 values of the compounds isolated from the plant.

Response 3: As per the reviewer’s suggestion, the information in Table 1 has been presented to summarize the various extracts’ anticancer activity against different types of cancer isolated from different parts of the plant, from leaves to fruits, seeds, barks, and branches. As stated previously many studies have reported the concentration-response effects of the extract as a whole but not of the individual compounds with precise IC50 values, limiting the information available to reported in a table. Moreover, IC50 values differ for different cell lines even for the same extract, which would be difficult to sort into a general table. Many of the biological compounds’ anticancer effects work in a synergistic way within the extract via multiple pathways, such as through their antioxidative, antiproliferative, and apoptotic activity. Focusing on the individual compounds isolated from the extracts would cause the manuscript to lose sight of the bigger picture.

Point 4: Figure 4. Mention the compounds that correspond to each activity, if they do not exist in the literature, place the extracts either from leaves, bark, roots, flowers or fruits.

Response 4: Table 2 has now been added in the revised manuscript to summarize the types of bioactive compounds that could be isolated from the different plant parts of P. guajava, and the probable mechanisms they utilize to inhibit the different types of cancer. In addition, one biomolecule can be isolated from the extracts made from different plant parts, and some studies used a combination of the plant parts such as leaves, bark, and root. Therefore, mentioning the compounds that corresponds to different activities as well as their origin of plant part would make the table too word-heavy and complicated.

Point 5: Conclusion

Based on the review carried out, mention which part of the plant has the greatest anticancer activity, and order them in order of activity.

Response 5: Based on the limited number of available studies, it is difficult to determine and rank order the part of the plant for their anticancer activity. Notably, even the same extract of one plant part works differently on different types of cancers with distinct effects (as evidenced by the IC50 or EC50 values of the same extract on different cell lines). Moreover, some extracts that have high antiproliferative effect against one type of cancer may not have an effect on the other.

Round 2

Reviewer 5 Report

All suggested changes were made. I suggest accepting the manuscript in its present form.